# Methionine Sulfoxide Reductases Suppress the Formation of the [*PSI*^+^] Prion and Protein Aggregation in Yeast

**DOI:** 10.3390/antiox12020401

**Published:** 2023-02-07

**Authors:** Jana Schepers, Zorana Carter, Paraskevi Kritsiligkou, Chris M. Grant

**Affiliations:** 1Institute of Pathobiochemistry, University Medical Center of the Johannes Gutenberg University Mainz, Duesbergweg 6, 55099 Mainz, Germany; 2Division of Molecular and Cellular Function, Faculty of Biology, Medicine and Health, The University of Manchester, Manchester M13 9PT, UK; 3Division of Redox Regulation, German Cancer Research Center (DKFZ), Im Neuenheimer Feld 280, 69120 Heidelberg, Germany

**Keywords:** prions, protein aggregation, methionine oxidation, methionine sulfoxide reductase, oxidative stress, yeast

## Abstract

Prions are self-propagating, misfolded forms of proteins associated with various neurodegenerative diseases in mammals and heritable traits in yeast. How prions form spontaneously into infectious amyloid-like structures without underlying genetic changes is poorly understood. Previous studies have suggested that methionine oxidation may underlie the switch from a soluble protein to the prion form. In this current study, we have examined the role of methionine sulfoxide reductases (MXRs) in protecting against de novo formation of the yeast [*PSI*^+^] prion, which is the amyloid form of the Sup35 translation termination factor. We show that [*PSI*^+^] formation is increased during normal and oxidative stress conditions in mutants lacking either one of the yeast MXRs (Mxr1, Mxr2), which protect against methionine oxidation by reducing the two epimers of methionine-S-sulfoxide. We have identified a methionine residue (Met124) in Sup35 that is important for prion formation, confirming that direct Sup35 oxidation causes [*PSI*^+^] prion formation. [*PSI*^+^] formation was less pronounced in mutants simultaneously lacking both MXR isoenzymes, and we show that the morphology and biophysical properties of protein aggregates are altered in this mutant. Taken together, our data indicate that methionine oxidation triggers spontaneous [*PSI*^+^] prion formation, which can be alleviated by methionine sulfoxide reductases.

## 1. Introduction

Methionine is a particularly oxidation-prone amino acid, forming a racemic mixture of methionine-*S*-sulfoxide and methionine-*R*-sulfoxide in cells [1]. The formation of methionine sulfoxide (MetO) can significantly influence protein structure and function via oxidation of the moderately hydrophobic thioester side chain on methionine into the hydrophilic sulfoxide form in MetO. Importantly, however, methionine oxidation can be reversed by the activity of methionine sulfoxide reductases (MXRs), which reduce MetO to methionine [2,3]. This means that reversible methionine oxidation can provide functional roles, including acting as an antioxidant in scavenging reactive oxygen species (ROS) [4,5,6]. This reversibility also means that methionine oxidation has increasingly become recognized as a reversible post-translational modification that regulates protein activity analogous to the action of protein phosphorylation [4,7,8,9].

The damaging consequences of MetO formation are evidenced by the many reports linking MXR activity with aging and the etiology of various neurological disorders [10,11,12,13,14]. Many neurodegenerative diseases in humans result from protein misfolding and aggregation, which can be triggered by methionine oxidation altering the protein structure/folding of normally soluble proteins. For example, methionine oxidation has been strongly implicated in prion formation. Prions are infectious misfolded proteins associated with fatal neurodegenerative diseases, including Creutzfeldt Jakob disease (CJD) in humans, scrapie in sheep, and bovine spongiform encephalopathy (BSE) in cattle [15]. Self-templating of the misfolded prion protein underlies the development of prion diseases in a mechanism that involves conversion of the normal prion protein (PrP) into its infectious PrP^Sc^ conformation [16,17]. Methionine oxidation has been identified during the spontaneous misfolding and aggregation of PrP during sporadic prion formation [18,19,20,21]. PrP misfolding and the formation of toxic oligomers and amyloid plaques may occur following methionine oxidation via PrP destabilization. Alternatively, methionine oxidation may cause PrP^SC^ stabilization, such as inhibiting the clearance of misfolded forms and favoring misfolding pathways, which result in amyloid formation. Many of these mammalian studies rely on in vitro approaches, and more evidence is required to establish a causal relationship between methionine oxidation and prion formation.

Numerous prion-like elements have now been described in fungi, including the yeast *Saccharomyces cerevisiae* [22,23,24]. The best-studied prion example is [*PSI*^+^], which is a self-propagating amyloid form of the Sup35 protein [22]. The Sup35 protein normally functions as a translation termination factor during protein synthesis, but the amyloid [*PSI*^+^] heritable prion form can be propagated in a protein-only mechanism. Spontaneous [*PSI*^+^] prion formation has long been known to be increased in response to *SUP35* overexpression, which increases the probability of prion formation [22]. Additionally, spontaneous [*PSI*^+^] prion formation is elevated in response to environmental stress conditions, including oxidative stress [25]. For example, the frequency of [*PSI*^+^] prion formation is increased following exposure to reactive oxygen species (ROS) such as hydrogen peroxide, implicating oxidative protein damage in the switch from a soluble to the heritable amyloid form of the protein [26,27,28].

The first indication that methionine oxidation might cause *de novo* [*PSI*^+^] prion formation came from the finding that oxidative stress conditions induced by exposure to hydrogen peroxide or the superoxide anion causes an increase in Sup35 MetO levels in parallel with an elevated frequency of [*PSI*^+^] prion formation [27,28]. Similar increases in Sup35 MetO levels were observed in antioxidant mutants, including mutants deficient in superoxide dismutases, catalases, and peroxiredoxins, which all display increased [*PSI^+^*] prion formation [27,28]. Increased MetO and prion formation is detected in antioxidant mutants grown under normal non-stress conditions, indicating that endogenous ROS levels are sufficient to promote Sup35 oxidation. Conversely, overexpression of MXR was found to protect Sup35 against methionine oxidation and suppressed [*PSI*^+^] prion formation in antioxidant mutants [27,28].

The correlation between MetO formation and [*PSI*^+^] prion formation suggests that methionine oxidation is linked with the switch from a soluble to an aggregated prion form of Sup35 during oxidative stress conditions. However, it is not known whether direct oxidative modification of Sup35 or, alternatively, oxidant-induced changes in the protein homeostasis network, promote prion formation. In this current study, we have examined the role of MXRs in protein aggregation and protection against [*PSI*^+^] prion formation. We show that global protein aggregation is increased in MXR mutants, indicating that MXRs are required to maintain protein homeostasis during normal non-stress conditions. The frequency of both oxidant and overexpression-induced prion formation is also increased in MXR mutants confirming that MXRs act to suppress amyloidogenic aggregation and prion formation. We have identified a methionine residue in Sup35 that is required for oxidant-induced [*PSI*^+^] formation, directly linking Sup35 methionine oxidation with prion formation.

## 2. Materials and Methods

### 2.1. Yeast Strains and Plasmids

The wild-type yeast strain 74D-694 (*MATa ade1-14 ura3-52 leu2-3112 trp1-289 his3-200* [*PIN*^+^][*psi*^−^]*)* was used for most experiments. Strains were deleted for *MXR1* (*mxr1::HIS3*) and *MXR2* (*mxr2::LEU2*) using standard yeast methodology. For comparison, *mxr1::KanMX* and *mxr2::KanMX* mutants were used in the BY4741 and BY4742 backgrounds from the Euroscarf yeast deletion collection [29]. Plasmids expressing fluorescently tagged proteins, including *HSP104-*RFP*, CUP1-SUP35NM-*GFP, and *GDP-HSP104*-mCherry, have been described previously [30,31,32].

Yeast strains expressing *SUP35-M124A* as the sole copy of *SUP35* were constructed using a plasmid shuffle approach described previously [33]. Briefly, a yeast strain deleted for the chromosomal copy of *SUP35* was complemented with a *URA3-CEN* plasmid carrying the wild-type *SUP35* gene. The *SUP35-M124A* mutant was constructed by cloning a commercially synthesized gene fragment bearing this mutation into plasmid pRS413 [34]. 5-Fluoro-orotic acid (5-FOA)-containing medium was used to select for cells expressing *SUP35-M124A* as the sole Sup35.

### 2.2. Growth and Stress Conditions

Yeast strains were grown at 30 °C with shaking at 180 rpm in minimal SD media (0.67% *w/v* yeast nitrogen base without amino acids, 2% *w/v* glucose) supplemented with appropriate amino acids and bases. Stress sensitivity was determined by growing cells to the stationary phase and spotting diluted cultures (A_600_ = 1.0, 0.1, 0.01) onto SD agar or YEPD plates containing various concentrations of hydrogen peroxide.

### 2.3. Protein and Western Blot Analysis

Protein extracts were electrophoresed under reducing conditions on SDS-PAGE minigels and electroblotted onto a PVDF membrane (Amersham Pharmacia Biotech). Protein concentrations were measured using a NanoDrop ND-8000 spectrophotometer. Primary antibodies used were Sup35 [35], GFP (Invitrogen, UK), Pgk1 (ThermoFisher Scientific, UK), and Hsp104 (Abcam, UK). Insoluble protein aggregates were isolated as previously described [36]. Aggregated proteins were visualized by silver staining with the Bio-Rad silver stain plus kit. The analysis of Sup35 amyloid polymers by semi-denaturing detergent agarose gel electrophoresis (SDD-AGE) was performed as described previously [37].

### 2.4. Protein Identification and Statistical Analysis

Aggregated proteins were identified by mass spectrometry (performed by the Biomolecular Analysis Core Facility, The University of Manchester) in triplicate for each condition. Protein samples were run a short distance into SDS-PAGE gels and stained using colloidal Coomassie blue (Sigma, UK). Total proteins were excised, and trypsin was digested and identified using liquid chromatography-mass spectrometry (LC-MS). Proteins were identified using the Mascot mass fingerprinting program (www.matrixscience.com accessed on 2 February 2023) to search the NCBInr and Swissprot databases. Final datasets for each condition were determined by selecting proteins that were identified in at least two of the three replicates. Venn diagrams and analysis of the distribution of protein hits between different strains were performed using Venny (http://bioinfogp.cnb.csic.es/tools/venny/ accessed on 2 February 2023). Mann–Whitney U-tests were used to assess the statistical significance of observed differences in hydrophobicity, abundance, translational efficiency, and protein size.

### 2.5. Fluorescence Microscopy

Cells were washed and immobilized on 10% poly-L-lysine-coated slides (Sigma-Aldrich, UK). All images were acquired on a Delta Vision (Applied Precision) restoration microscope using a 100×/NA 1.42 Plan Apo objective and fluorescein isothiocyanate (FITC) and Texas Red bandpass filters from the Sedat filter set (Chroma, USA). The images were collected using a Coolsnap HQ (Photometrics, USA) camera with a Z optical spacing of 0.2 μm. The images were then deconvolved using Softworx software and the maximum intensity or 3D projections. For display, images were processed and analyzed, including sizing of fluorescent puncta, using ImageJ [38]. Sup35 aggregation was visualized as described previously using *CUP1-SUP35NM-GFP* [27]. Cells were grown in SD minimal media and diluted to OD_600_ 0.1–0.2, and 50 μM copper sulfate was added for induction of the *CUP1* promoter and visualized.

### 2.6. Analysis of Prion Formation

Prion formation was quantified as described previously [39]. Yeast strain 74D-694 contains an assayable nonsense (UGA) mutation in the *ADE1* gene, which allows the frequency of [*PSI*^+^] prion formation to be scored by growth in the absence of adenine [39]. [*psi*^−^] strains are auxotrophic for adenine and appear red due to the accumulation of an intermediate in the adenine biosynthesis pathway. [*PSI*^+^] strains give rise to white/pink Ade^+^ colonies due to suppression of the *ade1-14* nonsense mutation and production of functional Ade1 protein. Diluted cell cultures were plated onto SD plates lacking adenine (SD-Ade) and incubated for 7–10 days. Prions were differentiated from nuclear gene mutations by their irreversible elimination on plates containing 4 mM GdnHCl. GdnHCl effectively blocks the propagation of yeast prions by inhibiting the key ATPase activity of Hsp104, a molecular chaperone that functions as a disaggregase and is required for prion propagation [40,41]. Data show the means of at least three independent biological repeat experiments expressed as the number of [*PSI*^+^] cells relative to viable cells. Data are presented as mean values ± SD.

## 3. Results

### 3.1. Strains Lacking Methionine Sulfoxide Reductases Are Unaffected in Hydrogen Peroxide Sensitivity

Yeast contains two MXRs: Mxr1 reduces the *S* stereoisomer of MetO and Mxr2 reduces the *R* stereoisomer of MetO [42]. We constructed *mxr1, mxr2,* and *mxr1,2* deletion mutants in a [*PIN*^+^][*psi*^−^] yeast strain (74D-694), which is commonly used to study yeast prion biology [43]. We first tested oxidant sensitivity by growing the wild-type and mutant strains in minimal media and spotting serial dilutions onto plates containing increasing concentrations of hydrogen peroxide. Surprisingly, no sensitivity to hydrogen peroxide was observed in any of the MXR mutants (Figure 1A). This lack of oxidative stress sensitivity is unexpected given previous reports that yeast MXR mutants are sensitive to oxidative stress induced by hydrogen peroxide [42].

This difference in sensitivity might arise due to differences in the yeast strain backgrounds used, particularly since the previous study used strains with a methionine auxotrophy, meaning that they cannot synthesize their own methionine, and oxidation of exogenously supplied methionine might influence sensitivity to oxidative stress. To test this possibility, the hydrogen peroxide sensitivity of MXR mutant strains in the BY4741 strain background, which is the methionine auxotroph (*met15Δ0*) used previously [42], was compared with the isogenic BY4742 strain background, which is prototrophic for methionine biosynthesis. In agreement with previous reports, the *mxr2* mutant and, to a lesser extent, the *mxr1* mutant strain were sensitive to oxidative stress conditions in the BY4741 background (Figure 1B). In comparison, loss of *MXR1* or *MXR2* did not cause hydrogen peroxide sensitivity in the BY4742 methionine prototrophic strain background. These findings indicate that methionine auxotrophy can influence oxidant sensitivity in MXR mutants. We, therefore, used the 74D-694 strain background to study the role of MXRs in proteostasis in the absence of any complications that might arise from methionine auxotrophy.

### 3.2. The Frequency of [PSI^+^] Prion Formation Is Increased in MXR Mutants

The frequency of [*PSI*^+^] formation was assayed in cells grown to exponential phase or exposed to oxidative stress conditions induced by exposure to 200 μM hydrogen peroxide for 20 h, which we have previously shown induces [*PSI^+^*] prion formation [28]. The measured frequency of [*PSI*^+^] prion formation was approximately 1 × 10^−6^ in the wild-type strain comparable to previously reported frequencies (Figure 2A) [44,45,46]. This was increased greater than 6-fold in the *mxr1* and *mxr2* mutants, suggesting that Mxr1 and Mxr2 normally act to suppress *de novo* [*PSI*^+^] formation caused by the presence of endogenous ROS (Figure 2A). Rather than having an additive effect, the frequency of [*PSI*^+^] formation was not increased in the double *mxr1,2* deletion strain compared with the wild-type strain. Following oxidative stress conditions, the frequency of [*PSI*^+^] formation was increased in all four strains. Significantly higher increases were observed in the single *mxr1* and *mxr2* mutant strains compared with the wild-type and double mutant strains (Figure 2A). Given that increased cellular concentrations of Sup35 can promote [*PSI*^+^] prion formation, the levels of Sup35 were measured in the MXR mutant strains. This analysis confirmed that similar levels of Sup35 are present in all strains, ruling out any effects on Sup35 protein concentrations (Figure 2B).

### 3.3. Mutation of Met124 in Sup35 Decreases the Frequency of [PSI^+^] Prion Formation

Elevated prion formation in MXR mutants might arise due to direct oxidation of Sup35, or alternatively, via oxidation and inactivation of chaperones or other components of the proteostasis defense machinery that normally suppress [*PSI*^+^] prion formation. We, therefore, attempted to directly test the role of Sup35 methionine residues in [*PSI*^+^] prion formation. Sup35 is an essential translation termination factor that contains three functional regions, including an N-terminal (N) region (residues 1–123) required for prion formation, a highly charged middle (M) region (residues 124–253), and a C-terminal region (residues 254–685) that is required for translation termination activity [47]. Sup35 contains a total of 19 Met residues, including its amino-terminal Met residue (Figure 2C). One Met residue is located at the end of the N domain (M124), and the other internal 17 Met residues are all located in the C-terminal catalytic region. We, therefore, took a focused approach by mutating M124 since we reasoned its location between the N and M domains makes it most likely to influence prion formation.

Strains containing an M124A substitution in Sup35 were found to be viable. However, the frequency of [*PSI*^+^] prion formation was reduced compared with the wild-type strain during non-stress conditions (Figure 2A). There was also no significant increase in the frequency of [*PSI*^+^] prion formation in response to oxidative stress in the M124A mutant, suggesting that Met124 is required for oxidant-induced prion formation. The elevated frequency of [*PSI*^+^] prion formation observed in the *mxr1* and *mxr2* mutant strains was also suppressed during both non-stress and oxidative stress conditions by mutation of Met124. Immunoblotting confirmed that the M124A version of Sup35 is expressed at similar levels to wild-type Sup35 (Figure 2C). Taken together, these data suggest that the Met124 residue in Sup35 is required to trigger oxidant-induced misfolding of Sup35, leading to [*PSI*^+^] prion formation.

### 3.4. Amorphous Protein Aggregation Is Increased in MXR Mutant Strains

Our data indicate that loss of Mxr1 or Mxr2 increases the frequency of [*PSI*^+^] formation, but there does not appear to be any additive effect due to the simultaneous loss of both MXRs. For comparison, we examined whether Mxr1 and Mxr2 are also required to protect against amorphous protein aggregation. Global protein aggregation was initially examined in MXR mutant strains using fluorescently tagged Hsp104. Hsp104 is the main cellular disaggregase that mediates refolding from the aggregated state [48] and can be used to visualize protein aggregate formation in cells [31,36,49]. Diffuse cytoplasmic Hsp104-RFP fluorescence was detected in most wild-type cells examined, and low levels of Hsp104-RFP puncta indicative of protein aggregate formation were detected in approximately 1% of cells (Figure 3A). The number of Hsp104-RFP puncta detected was similarly increased by approximately 10-fold in both the single and double MXR mutant strains, indicating that protein aggregation is elevated in MXR mutants.

A biochemical approach was next used where protein aggregates were isolated by centrifugation and membrane proteins removed using detergent washes [50,51,52,53]. The proteins in aggregates were separated using SDS-PAGE and visualized by silver staining. Low levels of protein aggregation were detected in the wild-type strain, and elevated protein aggregation was observed in the *mxr1*, *mxr2,* and *mxr1,2* mutants (Figure 3B). Again, no differences in the levels of protein aggregation were observed between the single and double mutant strains, suggesting that simultaneous loss of *MXR1* and *MXR2* does not have an additive effect on the levels of amorphous protein aggregation compared with single MXR mutants.

The proteins in the aggregated fractions were identified using mass spectrometry to determine whether specific types of protein aggregate in different MXR mutants. We identified 119 (wild-type), 165 (*mxr1*), 208 (*mxr2*), and 270 (*mxr1,2*) aggregated proteins (Appendix A). A higher number of proteins were identified in the MXR mutants compared with the wild-type strain, although pairwise comparisons indicated that many proteins commonly aggregated in both the wild-type and each MXR mutant (Figure 3C). Many unique proteins have also been identified that aggregate in each of the MXR mutants but not in the wild-type strain, and this was particularly apparent for the double *mxr1,2* mutant strain, where 163 aggregated proteins were identified that do not aggregate in the wild-type strain. Mxr1 primarily localizes in the cytosol, whereas, Mxr2 has been detected in mitochondria and cytosol [54,55]. We, therefore, examined whether there are any differences in the intracellular localization of the aggregated proteins in MXR mutants. We identified many aggregated proteins that are predicted to be mitochondrial proteins in the aggregated fractions from the wild-type and mutant strains. However, no enrichment for mitochondrial proteins was observed in the *mxr2* or *mxr1,2* mutants compared with the other strains (Figure 3D). It should be emphasized that we cannot say whether the aggregates themselves are formed within the mitochondria or cytosol since our aggregate preparations were made using whole-cell extracts. Most aggregated proteins were predicted to localize to the cytoplasm in all strains, along with localization in the nucleus, plasma membrane, ER, vacuole, and Golgi.

We assessed the physicochemical properties of the aggregated proteins to determine whether the proteins that aggregate in MXR mutants possess any properties that differentiate them. For this analysis, the aggregated protein datasets were compared with a list of yeast proteins detectable by mass spectrometry in a whole-cell extract, referred to as the MS set. Many studies have identified hydrophobicity as a driving force for protein aggregation [56,57,58]. In agreement with this idea, the aggregated proteins identified in the wild-type and MXR mutants were enriched for hydrophobic proteins compared with the MS set (Figure 4A). Protein abundance (molecules/cell) has also been shown to be a suitable indicator of aggregation propensity [59,60], and the aggregated proteins in the wild-type and MXR mutants were significantly enriched for abundant proteins compared with the MS set (Figure 4B). Consistent with this increase in abundance, the aggregated proteins were also enriched for proteins translated at higher rates, as determined using translational efficiency measurements [61] (Figure 4C). Interestingly, the aggregated proteins detected in the double *mxr1,2* mutant were less abundant (Figure 4B) and translated less efficiently (Figure 4C) compared with the aggregated proteins identified in the wild-type strain. The proteins that aggregate in the double *mxr1,2* mutant were also enriched for larger sizes compared with the wild-type and MS set (Figure 4D). Taken together, these data indicate that while aggregated proteins tend to be hydrophobic, highly abundant, and highly translated proteins, the overall threshold for protein aggregation (abundance, translation efficiency, and size) may be somewhat lower in the *mxr1,2* mutant, which may explain the higher number of aggregated proteins identified in this mutant compared with the wild-type and single mutant strains.

Finally, we compared the proteins that aggregate in MXR mutants with proteins that aggregate under different stress conditions (Figure 4E). Significant overlaps were identified with proteins that aggregate in response to oxidative stress caused by hydrogen peroxide, heat stress, and nascent protein-misfolding stress caused by the proline analog azetidine-2-carboxylic acid (AZC) [59,62]. This suggests that similar aggregation-prone proteins aggregate in MXR mutant strains and in response to different stress conditions. Yeast cells have been used as a model for the chronological lifespan of postmitotic cells, and we compared the aggregated proteins in MXR mutant stains with proteins that aggregate during yeast aging [63]. This analysis identified a significant overlap, suggesting that methionine oxidation may account for some of the aggregation that occurs during aging (Figure 4E). Taken together, these analyses indicate that protein aggregation is similarly increased in single and double MXR mutants, and common aggregation-prone proteins aggregate rather than proteins that are specifically affected in the absence of MXR activity.

### 3.5. Sup35 Aggregation Is Increased in MXR Mutants following Sup35 Overexpression

The best-established method to increase the frequency of *de novo* [*PSI*^+^] formation is to overexpress Sup35 [22]. This is because increasing the cellular concentrations of soluble Sup35 increases the probability of switching to the prion form. We, therefore, examined whether MXRs act to suppress overexpression-induced [*PSI*^+^] prion formation. For these experiments, we used a *SUP35NM-GFP* fusion protein that contains the amino-terminal prion-forming domain of Sup35 under the control of the copper-regulatable *CUP1* promoter to visualize Sup35 aggregate formation. One of the advantages of inducing [*PSI*^+^] formation via overexpression of *SUP35NM-GFP* is that the fluorescently tagged protein can be used to visualize Sup35 aggregation.

Overexpression of *SUP35NM-GFP* has been shown to result in the detection of fluorescent foci, which arise due to decorating existing aggregates, as well as rod- and ribbon-like aggregates characteristic of the *de novo* formation of [*PSI^+^*] [64,65,66,67]. We grew the wild-type, *mxr1, mxr2,* and *mxr1,2* mutants in the presence of copper to induce *SUP35NM-GFP* expression, and cells were visualized over a 20-hour time course (4, 8, 16, and 20 h). Puncta formation was first observed at low frequencies (<2% of cells) in the wild-type strain following overexpression of *SUP35NM-GFP* for 16 h, which increased to approximately 5% of cells following 20 h (Figure 5A). In contrast, visible puncta were observed as early as the four-hour time point in the *mxr1* (~9% cells), *mxr2* (~17% cells), and *mxr1,2* (~1% of cells) mutant strains, and more puncta were detected in the MXR mutant strains at all time points compared with the wild-type strain. Interestingly, ribbon and ring formation was observed at all time points in the *mxr1* and *mxr2* mutants, and this accounted for a large proportion of the fluorescent aggregates detected in the single mutants after 20 h overexpression of *SUP35NM-GFP* (Figure 5A). No ribbon or ring formation was detected in the *mxr1,2* mutant at any time point.

Initial visual inspection of the aggregates formed in the *mxr1,2* double mutant suggested that the aggregates are larger and less well-defined compared with the aggregates visualized in the wild-type and single MXR mutant strains (Figure 5A). To quantify this potential difference, GFP-positive areas were measured in the various strains following induction of *SUP35NM-GFP* for 20 h (Figure 5B). The average sizes of the GFP-positive foci in the wild-type, *mxr1,* and *mxr2* mutant strains were comparable, whereas the average GFP-positive area was significantly larger in the *mxr1,2* mutant strain. This suggests that the morphology of the aggregates formed in the double deletion strain is different from that of the aggregates formed in the single deletion strains.

We used semi-denaturing detergent agarose gel electrophoresis (SDD-AGE) to compare the Sup35 aggregates formed in the wild-type and MXR mutant strains. SDD-AGE takes advantage of the SDS-resistance of amyloid aggregates to facilitate diagnostic visualization of the high-molecular-weight amyloid aggregates formed in cells [68]. Western blotting was used to examine Sup35, which was detected in its monomeric form in a control [*psi*^−^] strain and in both its monomeric and high-molecular-weight SDS-resistant forms in a control [*PSI*^+^] strain (Figure 5C). A very similar pattern of Sup35 aggregation was observed in the wild-type, *mxr1,* and *mxr2* mutant strains following overexpression of *SUP35NM-GFP* for 20 h. In contrast, an increase in high-molecular-weight SDS-resistant Sup35 aggregates was detected in the *mxr1,2* double deletion, and these encompassed a wider range of molecular weights compared with the other strains (Figure 5C).

### 3.6. Overexpression-Induced [PSI^+^] Prion Formation Is Increased in Single but Not Double MXR Mutant Strains

The formation of Sup35 aggregates does not necessarily indicate heritable [*PSI^+^*] prion formation. We, therefore, quantified the frequency of [*PSI*^+^] formation following overexpression of *SUP35NM-GFP* for 20 h in the wild-type, *mxr1, mxr2,* and *mxr1,2* mutant strains. As expected, the frequency of [*PSI*^+^] prion formation was strongly induced in the wild-type strain reaching a frequency of 3 × 10^2^ (Figure 6A). [*PSI*^+^] prion formation was increased a further four- to five-fold in the *mxr1* and *mxr2* mutant strains compared with the wild-type strain. In contrast, the frequency of overexpression-induced [*PSI*^+^] formation in the *mxr1,2* double mutant was comparable to the wild-type (Figure 6A). This is despite a greater than two-fold increase in the frequency of Sup35 aggregates formed in the *mxr1,2* double mutant compared with the wild-type strain (Figure 5A), suggesting that many of the Sup35 aggregates formed in the *mxr1,2* mutant are non-productive in terms of heritable prion formation.

One possible explanation for the larger Sup35 aggregates and the reduced frequency of prion formation in the *mxr1,2* mutant is that the aggregates are poor substrates for Hsp104 disaggregase activity. Previous studies have shown that decreases in Hsp104 activity result in an accumulation of larger Sup35 aggregates that limit the transmission of prions into daughter cells [43,68,69]. We, therefore, examined whether overexpression of Hsp104 could disaggregate the large Sup35 aggregates detected in the *mxr1,2* mutant strain using strains expressing Hsp104 under the control of the constitutively active *TDH3* promoter [30]. Western blotting was first used to confirm that Sup35 and Hsp104 were similarly overexpressed in all strains (Figure 6B).

Overexpression of Hsp104 was found to increase the number of *mxr1,2* mutant cells containing Sup35 puncta to greater than 16% of cells (Figure 6C). Interestingly, ribbon and ring formation was also now detected in the *mxr1,2* mutant. In comparison, overexpression of Hsp104 decreased the formation of ring and ribbon aggregates in the single *mxr1* and *mxr2* mutant strains (Figure 6C). Microscopy analysis of puncta formation confirmed that smaller aggregates were detected in the *mxr1,2* double mutant, and similar-sized Sup35 aggregates were detected in all strains examined following Hsp104-overexpression (Figure 6D). Overexpression of Hsp104 is known to cure yeast cells of the [*PSI*^+^] prion, presumably due to the solubilization of Sup35 protein aggregates [70]. Similarly, overexpression of Hsp104 was found to decrease the frequency of [*PSI*^+^] prion formation in the single and double MXR mutant strains (Figure 6A). These data indicate that an inability of Hsp104 to solubilize the larger Sup35 aggregates in the *mxr1,2* mutant strain does not appear to explain the reduced frequency of [*PSI*^+^] prion formation in the double compared with the single MXR mutant strains.

## 4. Conclusions

Protein aggregation is the abnormal association of misfolded proteins into larger, often insoluble structures whose level can be increased by many genetic, cellular, and environmental factors, including advanced age [71,72]. Newly synthesized proteins are particularly vulnerable to misfolding and aggregation arising from translational errors occurring during their synthesis and from mistakes made while folding into their final quaternary structures [73]. This can be exacerbated by environmental stress conditions that increase protein unfolding, leading to aggregation [74]. Protein aggregation is generally classified into two categories: amyloid and amorphous. The amyloid state is a highly structured, insoluble fibrillar deposit, usually consisting of many repeats of the same protein. Fibrillar assemblies of prion-like proteins are implicated in age-related neurodegenerative diseases, including Alzheimer’s, Parkinson’s, and Huntington’s. Amorphous aggregation is the disordered aggregation of proteins into aggregates without forming a specific higher-order structure. We found that MXR activity is required to suppress the formation of both types of aggregation, suggesting that methionine oxidation underlies widespread protein aggregation in cells.

MXRs are well-known antioxidants that are required to reduce oxidized methionine residues, thereby protecting proteins against oxidative damage [2]. Similar increases in the levels of protein aggregation were detected using two different approaches, namely biochemical purification of high-molecular-weight aggregates and using the Hsp104 chaperone to decorate the sites of aggregation in cells. This suggests that MXRs play an active role in maintaining proteostasis during normal, non-stress growth conditions. Methionine oxidation has long been known to cause protein misfolding and destabilization that can ultimately reduce the activity of enzymes and other proteins [2,75,76,77]. Not all methionine residues are susceptible to oxidation, and the oxidation of different methionines within the same protein can vary depending on their localized environments and surface exposure [78,79,80,81]. The extent of protein aggregation as well as the identification of proteins with similar biophysical properties in *mxr1* and *mxr2* mutant strains, suggests that the chirality of methionine oxidation does not have any differential effects on protein aggregation in yeast. The properties of the aggregated proteins identified in MXR mutants were also typical for aggregation-prone proteins identified in other studies, including high abundance and high hydrophobicity [59]. Despite the similar levels of protein aggregation detected, a greater number of aggregated proteins were identified in the *mxr1,2* double mutant strain compared with the single MXR mutants. The aggregated proteins detected in the *mxr1,2* mutant were also less abundant, less well-translated, and larger than the aggregated proteins identified in the single MXR mutants, suggesting that the threshold for protein aggregation may be somewhat lower in mutants lacking all MXR activity. However, despite this apparent reduction in the threshold for aggregation, cells appear to have mechanisms to prevent aggregates from further accumulating in the double *mxr1,2* mutant.

Loss of Mxr1 or Mxr2 was found to increase the frequency of [*PSI*^+^] prion formation, consistent with the idea that under normal growth conditions, MXRs are sufficient to suppress prion formation during exposure to endogenous ROS. The finding that prion formation was similarly increased in *mxr1* and *mxr2* mutants suggests that both the *S* and *R* stereoisomers of MetO can promote misfolding underlying the switch from the soluble to the prion form of Sup35. Given the differences in the substrate specificity of Mxr1 and Mxr2, it was surprising that there was no additional increase in the frequency of [*PSI*^+^] prion formation in the double mutant, and prion formation was more comparable to the wild-type strain. Limited data are available regarding the chirality of methionine oxidation occurring in live cells, but given that MXRs specific for both epimers can influence tolerance to oxidative stress in multiple organisms, it seems likely that both forms of MetO must form in cells [7,82,83]. Oxidative stress conditions further increased the frequency of *de novo* prion formation in wild-type and MXR mutant strains, but again the frequency of [*PSI*^+^] formation was less elevated in a double *mxr1,2* mutant in response to oxidative stress conditions compared with the single MXRs mutants. It is unclear why the simultaneous loss of Mxr1/2 might decrease prion formation compared with single mutant strains. One possibility is that larger Sup35 aggregates may be formed in the double mutant during oxidative stress conditions compared to the large aggregates formed in response to the overexpression of Sup35. As discussed below, smaller Sup35 aggregates are more efficiently transmitted to daughter cells to propagate [*PSI*^+^] formation, and hence the nature of the Sup35 protein aggregates formed can profoundly influence the efficiency of prion propagation [68,69].

Increased [*PSI*^+^] prion formation might arise in MXR mutant strains due to the oxidation of Sup35 methionine residues or oxidation and inactivation of the normal protein homeostasis machinery that maintains Sup35 in its soluble form [19]. Interestingly therefore, we found that Met124 in Sup35 is required for oxidant-induced prion formation in response to hydrogen peroxide stress and for the increased frequency of [*PSI*^+^] prion formation observed in MXR mutants, suggesting that direct Sup35 protein oxidation promotes prion formation. This is analogous to mammalian PrP, where methionine oxidation is thought to destabilize the native protein facilitating misfolding to form the amyloid PrP^sc^ form [20]. Sup35 methionine oxidation could cause misfolding by altering the conformation of its polypeptide backbone and decreasing its thermal stability, or alternatively, it may disrupt normal Sup35 interactions with the protein homeostasis machinery. Yeast contains many chaperone and anti-prion systems that act to suppress prion formation [84], and hence Sup35 oxidation may alter protein–protein interactions with these systems. Whatever the mechanism, it is clear that MXRs normally act as protective systems that suppress prion formation during exposure to both endogenous and exogenous sources of ROS.

A key genetic criterion that defines yeast prions is that overexpression of the normally soluble protein results in an increased frequency of *de novo* prion formation [85]. Interestingly, therefore, overexpression-induced [*PSI*^+^] formation was found to be significantly higher in strains lacking *MXR1* or *MXR2* compared with the wild-type strain. Sup35 was similarly overexpressed in all strains, indicating that overexpression and methionine oxidation have additive effects on prion formation. It is unknown whether Sup35 oxidation occurs on the nascent polypeptide chain or in pre-existing Sup35 molecules. The kinetics of Sup35 protein aggregation and the formation of the ring and ribbon-like structures indicative of *de novo* prion formation [64,65,66,67] was faster in the *mxr1* and *mxr2* mutants, suggesting that newly expressed Sup35 rapidly misfolds and aggregates in the absence of Mxr1 or Mxr2 activity.

The kinetics and frequency of Sup35 aggregation were reduced in the double *mxr1,2* mutant compared with the single MXR mutants, suggesting that different aggregates are formed in this mutant. The increase in the size of the Sup35 aggregates and the higher-molecular-weight SDS-resistant aggregates formed in the double deletion strain may explain the lower frequency of prion formation compared with the single deletion strains. This is because smaller Sup35 aggregates are thought to be more efficiently transmitted to daughter cells to propagate [*PSI*^+^] formation [68,69]. This raises the question as to why larger Sup35 aggregates accumulate in the double *mxr1,2* strain compared with the wild-type and single MXR mutants. The Hsp104 disaggregase is essential for yeast prion propagation [40,41,69]. Given that Hsp104 is unlikely to be limiting in the *mxr1,2* double mutant based on the similar number of Hsp104-marked protein aggregates detected in the single and double MXR mutants, one possibility is that the larger Sup35 aggregates might be poor substrates for Hsp104 activity. However, overexpression of Hsp104 was found to result in similar-sized Sup35 aggregates being formed in the double MXR compared with the other strains, suggesting that the Hsp104 disaggregase is active against the larger Sup35 aggregates formed in the double *mxr1,2* mutant. Cells contain an arsenal of molecular chaperones that maintain proteostasis, and it is possible that another chaperone that normally suppresses prion formation is particularly inactivated in the complete absence of MXR activity. For example, oxidation of methionine residues in the Fes1 co-chaperone inhibits its activity to modulate Hsp70 activity, which is thought to be important during oxidative stress conditions [55].

Further work will be required to define how the MetO/MXR-coupled system affects the proteostasis machinery, but our finding that global protein aggregation levels are elevated in MXR mutants reinforces the role of MXRs in protecting the proteome against oxidative damage and protein aggregation. Methionine oxidation appears to play a key role in *de novo* yeast prion formation, and it will be important to further investigate its influence in the formation of other sporadic protein-misfolding events, many of which underly neurological disorders such as Alzheimer’s, Parkinson’s, and Huntington’s.

## Figures and Tables

**Figure 1 antioxidants-12-00401-f001:**
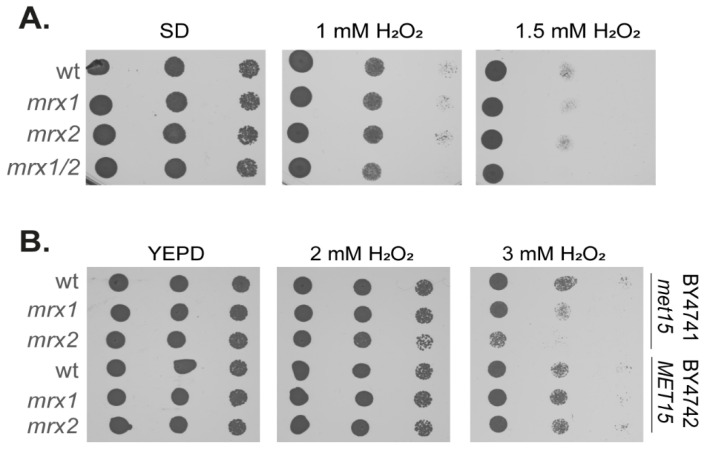
Strains lacking MXRs are unaffected by hydrogen peroxide sensitivity. (**A**) Wild-type (74D-694) and MXR mutant strains were grown to the exponential phase, and the A_600_ adjusted to 1, 0.1, 0.01, or 0.001 before spotting onto plates containing the indicated concentrations of hydrogen peroxide. (**B**) Wild-type (BY4741, BY4742) and isogenic MXR mutant strains were tested for hydrogen peroxide sensitivity as described in panel (**A**).

**Figure 2 antioxidants-12-00401-f002:**
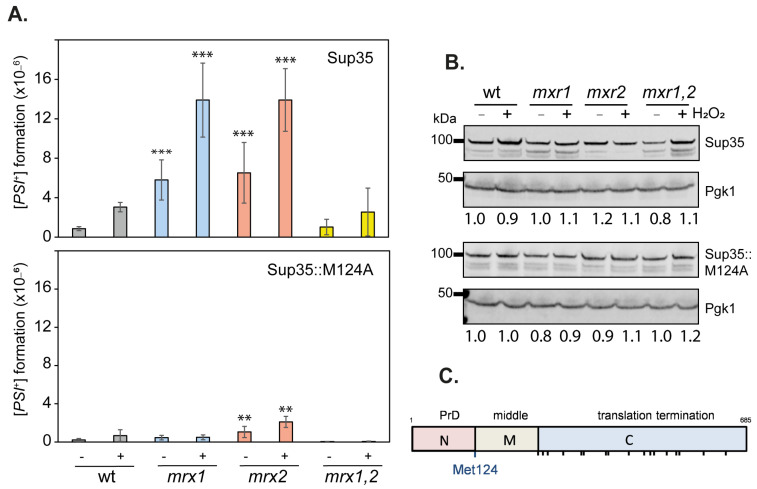
The frequency of [*PSI*^+^] prion formation is increased in single MXR mutants. (**A**) [*PSI*^+^] prion formation was quantified in the wild-type and MXR mutant strains in the presence or absence of hydrogen peroxide. This was repeated in strains containing wild-type Sup35 or the M124A mutant. Data shown are the means of three independent biological repeat experiments expressed as the number of colonies per viable cell. Error bars denote standard deviation. Significance (one-way ANOVA) is shown compared with the wild-type strain in the absence or presence of hydrogen peroxide; ** *p* < 0.01, *** *p* < 0.01. (**B**) Western blot analysis of the same strains probed with αSup35 or α-Pgk1 as a loading control. Band intensities were quantified and are shown comparing Sup35 with Pgk1 normalized to the untreated wild-type strains (lane 1). (**C**) Schematic showing domain structure of Sup35. The Sup35 protein can be divided into 3 distinct regions: an N-terminal prion-forming domain (PrD), a highly charged middle region (M), and a C-terminal domain that functions in translation termination (**C**). Sup35 contains a total of 19 Met residues, including its amino-terminal Met residue. One Met residue is located between the N and M regions (M124), and the other internal 17 Met residues are all located in the C-terminal catalytic region.

**Figure 3 antioxidants-12-00401-f003:**
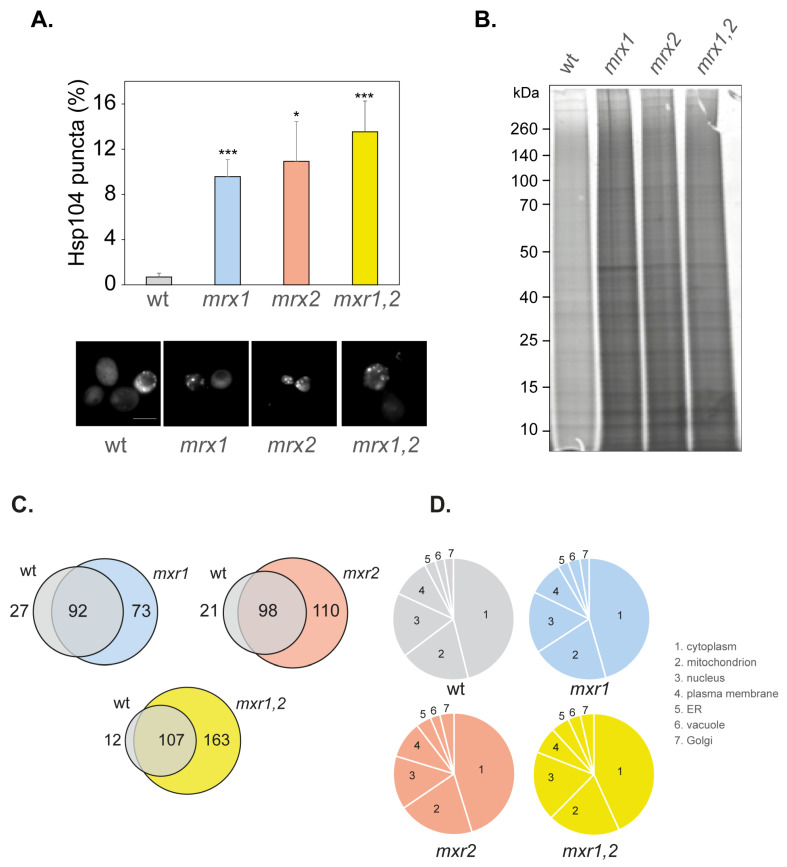
Strains lacking MXRs have higher levels of protein aggregation. (**A**) Hsp104-RFP was visualized in wild-type and MXR mutant cells grown to the exponential phase. Examples of cells containing visible puncta are shown. The percentage of cells containing visible Hsp104-RFP puncta was quantified for each strain. Data shown are the means of three independent biological repeat experiments ± SD. Significance is shown compared with the wild-type strain; * *p* < 0.05, *** *p* < 0.001. (**B**) Protein aggregates were isolated from the same strains and analyzed by SDS-PAGE and silver staining. (**C**) Proteins within insoluble aggregate fractions were identified by mass spectrometry. The Venn diagrams show pairwise comparisons of the overlaps between the proteins aggregating in wild-type and MXR mutant strains (*mxr1*, *mxr2, mxr1,2*). (**D**) Venn diagrams showing the localization of the proteins that aggregate in the wild-type and MXR mutants.

**Figure 4 antioxidants-12-00401-f004:**
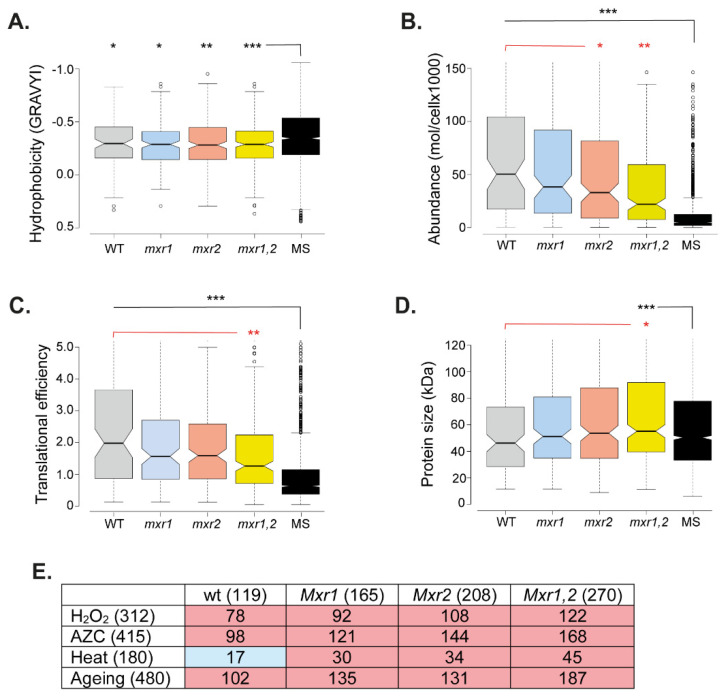
The localization and physicochemical properties of aggregated proteins are similar in wild-type and MXR mutants. Comparison of the aggregated proteins present in MXR mutants and the wild-type strain. Aggregated proteins were compared with a list of unaggregated proteins identified by mass spectrometry, referred to as the MS set. (**A**) Grand average of hydrophobicity (GRAVY). (**B**) The abundance of proteins (molecules/cell) in each set during non-stress conditions. (**C**) Translational efficiency (TE) expressed as the ratio of ribosome footprint density to mRNA density [61]. (**D**) Protein size (kDa). Mann–Whitney U-tests were used to assess the statistical significance of observed differences: * *p* < 0.05, ** *p* < 0.01, *** *p* < 0.001. (**E**) Comparison of the aggregated proteins present in the wild-type and MXR mutants with other datasets including proteins that aggregate during heat-shock [62], proteins that aggregate during AZC stress [59], proteins that aggregate during hydrogen peroxide stress [59], proteins that ag-gregate during postmitotic ageing in yeast [63]. Numbers in brackets indicate the size of each dataset. The significance of overlaps was determined by a hypergeometric test. Blue: *p* < 0.01, red: *p* < 0.001.

**Figure 5 antioxidants-12-00401-f005:**
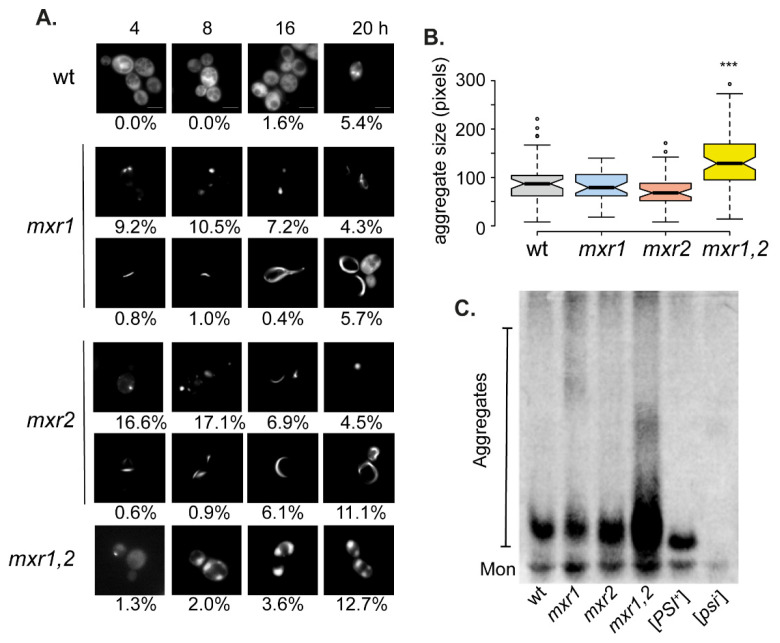
Visualization of overexpression-induced aggregation of Sup35. (**A**) Fluorescence micrographs are shown for wild-type and MXR mutant strains containing the Sup35NM-GFP plasmid induced with copper for the indicated times. Representative images are shown for puncta or rod- and ribbon-like aggregates at the indicated time points. The percentage of cells containing visible puncta or rod- and ribbon-like aggregates is indicated for at least 300 cells counted for each strain. (**B**) The GFP-positive areas were quantified for puncta formed in each strain following induction of Sup35NM-GFP for 20 h. At least 38 puncta were measured from each strain, and statistical significance was determined using Mann–Whitney U-tests (*** *p* < 0.001). (**C**) SDS-resistant Sup35 aggregates were detected in the same strains using SDD-AGE. [*PSI*^+^] and [*psi*^−^] derivatives of 74D-694 are shown for comparison.

**Figure 6 antioxidants-12-00401-f006:**
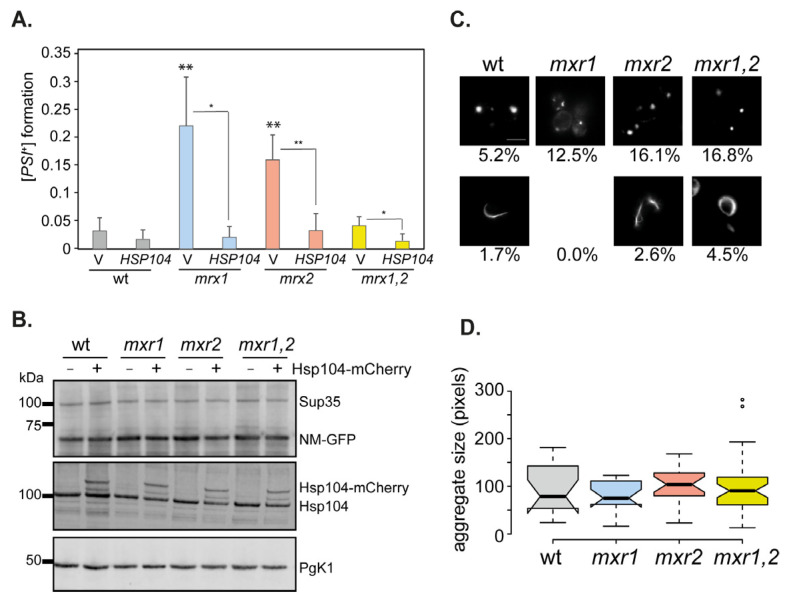
Overexpression-induced [*PSI*^+^] prion formation is increased in single but not double MXR mutant strains. (**A**) [*PSI*^+^] prion formation was quantified in the wild-type and MXR mutant strains containing vector alone or expressing Hsp104 under the control of the constitutively active *TDH3* promoter. Prion formation was induced by inducing Sup35NM-GFP expression following 20 h of copper induction. Data shown are the means of three independent biological repeat experiments expressed as the number of [*PSI*^+^] cells per viable cell. Error bars denote standard deviation. Significance was tested using an unpaired t-test, and pairwise comparisons are for the wild-type strain and MXR mutants containing the vector, or each strain comparing vector with *HSP104;* * *p* < 0.05, ** *p* < 0.01. (**B**) Western blot analysis of the same strains probed with αSup35 (endogenous Sup35 and NM-GFP), αHsp104 (endogenous Hsp104 and Hsp104-mCherry), or α-Pgk1 as a loading control. (**C**) Fluorescence micrographs showing examples of puncta or rod- and ribbon-like aggregates formed in the wild-type and MXR mutant strains expressing Hsp104 under the control of the constitutively active *TDH3* promoter. Prion formation was induced by inducing Sup35NM-GFP expression following 20 h of copper induction. The percentage of cells containing visible puncta or rod- and ribbon-like aggregates is indicated. (**D**) The GFP-positive areas were quantified for puncta formed in panel (**C**). At least 16 puncta were measured from each strain.

## Data Availability

Data is contained within the article or Appendix A.

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
