# Peer review of "Methionine Sulfoxide Reductases Suppress the Formation of the [PSI+] Prion and Protein Aggregation in Yeast"

_antioxidants, 2023, doi:10.3390/antiox12020401_

Round 1
Reviewer 1 Report
Yeast cells can form insoluble aggregates. Aggregate formation can be induced by prion proteins such as Sup35 or Rnq1. These aggresome- or IPOD-type of aggregates are often toxic. In addition, small heat shock proteins such as Hsp42 and Hsp26 can serve as nucleation seeds for aggregates. The latter type of aggregates is also referred to as JUNQ or Q bodies, is more benign and can be resolved once stress conditions are over. Both types of aggregates are bound be the disaggregase Hsp104. In the current study, the authors elucidated the relevance of enzymes that counteract methionine oxidation for Sup35 aggregation (first part of the study). In addition, they analyze the proteome of aggregates in such cells in general without further differentiation of the nature of these aggregates. Finally, they monitor the distribution of Hsp104 in these cells.
Interestingly, they identify a single methionine residue in Sup35 as to be relevant for its aggregation behavior. The client spectrum identified by proteomics shows many mitochondrial proteins and it remains unclear whether they are part of intra-mitochondrial or cytosolic aggregates. The proteomics part should be better analyzed and visualized. In general, this is an interesting study which needs some minor amendments.
Specific points:
1. The analysis of the proteomics experiment should be improved. The authors should provide volcano plots to compare aggregated proteins of (i) wild type vs. mrx1 deletion cells and (ii) mrx1 deletion vs. mrx2 deletion cells. Thereby enrichment scores (log fold changes) and q values for statistical significance should be plotted.
2. Excel tables need to be included into the supplement showing the results of the proteomics experiment.
3. The authors should show a PCA analysis of the result. They claim that there is no considerable difference between the aggregates in mrx1 and mrx2 mutants. A PCA analysis or a heat plot would demonstrate this more clearly.
4. Fig. 2A: It is not clear to me why the double deletion of mrx1 and mrx2 wipes out Sup35 aggregation. Even if the effect of the methionine oxidases is not additive, one would expect similar levels than in a single mutant. This should be better explained.
5. The authors identified many mitochondrial proteins in the aggregates. Are these aggregates within mitochondria or in the cytosol? This should at least be discussed.
6. Lane 202: ‚exposure to 200 M hydrogen peroxide’. Please correct the unit, 200 M seems to be a lot!
7. Lane 377 needs to be corrected
Author Response
Reviewer 1
Specific points:
- The analysis of the proteomics experiment should be improved. The authors should provide volcano plots to compare aggregated proteins of (i) wild type vs. mrx1 deletion cells and (ii) mrx1 deletion vs. mrx2 deletion cells. Thereby enrichment scores (log fold changes) and q values for statistical significance should be plotted.
We cannot provide volcano plots as requested by the reviewer since the analysis is not quantitative. As detailed in the Methods, aggregated proteins were identified by mass spectrometry in triplicate experiments for each condition. Final datasets were then determined by selecting proteins that were identified in at least two of the three replicates. Our analysis is therefore based on proteins being identified as being present in aggregate fractions rather than quantifying their cellular distribution.
- Excel tables need to be included into the supplement showing the results of the proteomics experiment.
We have added Supplementary Table 1 which is an Excel Table listing the aggregated proteins identified in each strain as defined above.
- The authors should show a PCA analysis of the result. They claim that there is no considerable difference between the aggregates in mrx1 and mrx2 mutants. A PCA analysis or a heat plot would demonstrate this more clearly.
As detailed in response to point 1, we cannot provide a PCA analysis or heat map as this is not a quantitative analysis.
- 2A: It is not clear to me why the double deletion of mrx1 and mrx2 wipes out Sup35 aggregation. Even if the effect of the methionine oxidases is not additive, one would expect similar levels than in a single mutant. This should be better explained.
We agree with the reviewer that it is surprising that the frequency of prion formation is reduced in the single mutants compared with the double mutant. We think that this is particularly interesting and have expanded the discussion on lines 527-534 of the revised manuscript.
- The authors identified many mitochondrial proteins in the aggregates. Are these aggregates within mitochondria or in the cytosol? This should at least be discussed.
As indicated by the Reviewer, we identified many aggregated proteins which are predicted to be mitochondrial proteins in the aggregated fractions from all strains. However, we cannot say whether the aggregates themselves are within the mitochondria or cytosol since our aggregate preparations were made using whole cell extracts. We have clarified this on pages 302-309 of the revised manuscript.
- Lane 202: ‚exposure to 200 M hydrogen peroxide’. Please correct the unit, 200 M seems to be a lot!
The typo has been corrected.
- Lane 377 needs to be corrected
The formatting error has been corrected
Reviewer 2 Report
This review is written by F. Caudron.
In this article, Schepers et al. address the role of methionine oxidation in the induction of the prion form of the Sup35 protein. Sup35 is the most described prion protein in yeast and highly influences the field of yeast prion biology, and prion biology in general. Here, the authors used strains deleted for the methionine sulfoxide reductases to test the role of these enzymes in prion conversion. The authors show that Sup35 prion conversion is increased in the single mutants mrx1∆ and mrx2∆. They also show that this induction depends on the Hsp104 disaggregase. In addition, the authors identify a methionine in the sequence of Sup35 (Met124), that is required for prion induction. This result strengthens the conclusion that methionine oxidation of the Sup35 protein can directly induce prion conversion. Accordingly, prion induction in mrx1∆ and mrx2∆ single mutants is suppressed by a mutation of Met124.
The double mutant mrx1∆ mrx2∆ is more puzzling as prion conversion is not increased in these cells. This may come from the observation that proteostasis is clearly affected in these cells and that the Sup35 assemblies appear very different from the other conditions, as shown by fluorescence microscopy and biochemical analysis.
Finally, the authors show that the increased sensitivity to H2O2 of the single mrx1∆ or mrx2∆ cells is only apparent in met15∆ cells, hence result from a synthetic interaction of these genes. However, it seems from Fig1A that the double mutant mrx1∆ mrx2∆ has a mild sensitivity (the third dilution does not grow), which is not described in the text. Regarding this experiment, it could also be that the sensitivity may be different if cells were grown exponentially before plating, instead of driving them to stationary phase. Could the authors comment on this?
Overall, the article is well written, complete (except one part, see major point 2) and adds to our understanding of prion conversion in the cell.
Major points:
1. Fig 2B: it seems that Sup35 levels are increased by H2O2 in WT and mrx1∆ mrx2∆ cells, not influenced by MRX1 deletion and decreased by MRX2 deletion, while the authors conclude that the levels of Sup35 are unchanged. It would be good to provide a quantification of these results, particularly if the western blots have been repeated.
2. To my point of view, the list of proteins identified in the mass spectrometry analysis of aggregated protein in the four strains should be provided to the reader.
Minor points:
1. line 202: I suppose the character µ (micron) is missing (200 µM)
2. line 377: the sentence in this line is a continuation of the sentence of line 368.
I suppose both points 1 and 2 arose during the poor conversion to pdf that the submission system of mdpi does. This needs to be fixed by mdpi.
3. I would suggest that the authors use an inverted LUT to present the pictures of fluorescence imaging.
Author Response
Reviewer 2
Major points:
- Fig 2B: it seems that Sup35 levels are increased by H2O2 in WT and mrx1∆ mrx2∆cells, not influenced by MRX1 deletion and decreased by MRX2 deletion, while the authors conclude that the levels of Sup35 are unchanged. It would be good to provide a quantification of these results, particularly if the western blots have been repeated.
Quantification for Sup35 and Pgk1 is now provided on Fig. 2B. These data show that the relative concentrations of Sup35 do not differ by greater than 20% for the wild-type and mutant strains in the presence or absence of hydrogen peroxide.
- To my point of view, the list of proteins identified in the mass spectrometry analysis of aggregated protein in the four strains should be provided to the reader.
Supplementary Table 1 provides an Excel Table listing the aggregated proteins identified in each strain.
Minor points:
- line 202: I suppose the character µ (micron) is missing (200 µM)
The typo has been corrected.
- line 377: the sentence in this line is a continuation of the sentence of line 368.
Corrected
- I would suggest that the authors use an inverted LUT to present the pictures of fluorescence imaging.
We prefer to keep the fluorescent images in their current format since it is consistent with our previous publications and others in the literature showing the patterns of Sup35 protein aggregation in cells.